# An Environmentally Friendly Supramolecular Glue Developed from Natural 3,4-Dihydroxybenzaldehyde

**DOI:** 10.3390/polym14050916

**Published:** 2022-02-25

**Authors:** Hui Wang, Xin Du, Yuanyuan Liu, Xingjiang Liu, Ailing Sun, Liuhe Wei, Yuhan Li

**Affiliations:** College of Chemistry and Green Catalysis Center, Zhengzhou Key Laboratory of Elastic Sealing Materials, Zhengzhou University, Zhengzhou 450001, China; h18839782096@163.com (H.W.); duxin_1911@163.com (X.D.); yuanyuanliu11@163.com (Y.L.); xingjiangliu@zzu.edu.cn (X.L.); ailingsun@zzu.edu.cn (A.S.)

**Keywords:** polymer gels, coordination, 3,4-dihydroxybenzaldehyde, environmentally friendly, adhesion

## Abstract

Liquid adhesive suffers from the emission of volatile organic compounds (VOCs) that have detrimental effects on human beings. Herein, an environmentally friendly glue containing a novel supramolecule dissolved in non-toxic ethanol is developed. Poly (ether amine) (PEA) and 3,4-dihydroxybenzaldehyde (dhba) is utilized to synthesize catechol-terminated PEA, and subsequent complexation by Fe^3+^ results in the supramolecular component (PEA-dhba-Fe^3+^). The Fourier transform infrared (FTIR) spectrum together with the UV-vis spectrum reveal the existence of quinone converted from catechol. Raman spectra prove the existence of a successful complex of catechol-terminated PEA with Fe^3+^. The tri-complex is found to be the predominant mode and can successfully form into clusters, serving as a physical cross-linking network. The PEA-dhba-Fe^3+^ exhibits strong adherence to metal substrates compared to polymeric substrates, with its shear strength reaching as high as 1.36 ± 0.14 MPa when the pH of the glue is adjusted to 8. The obvious improvement of adhesion originates from the formation of interfacial coordination bonds between quinone/catechol and metal atoms, as well as their cations, as revealed by X-ray photoelectron spectroscopy (XPS) and theoretical calculations. With consideration of its merits, including strong adhesion and the minor emission of VOCs compared to commercial epoxy and acrylic adhesives, this environmentally friendly supramolecular glue has a range of cutting-edge applications as an adhesive for metal substrates.

## 1. Introduction

Marine mussels have the ability to adhere to the surface of foreign bodies in seawater by secreting adhesion proteins. These mussel foot proteins (Mfps) are known to cure rapidly to form adhesive plaques with high interfacial binding strength, durability, and toughness. Through this process, mussels are able to firmly bind with organic and inorganic surfaces in aqueous environments where many other glues fail [1,2]. Herein, 3,4-dihydroxyphenylalanine (dopa), which is hydroxylated by tyrosine through the transition, is one of the main components of Mfps [3,4]. Moreover, it was recently demonstrated that a small amount of Fe^3+^ (<1 wt%) plays an important role in this mechanism by bonding with the catechol-like amino acid (dopa) in the cuticle protein, mfp-1. Upon the addition of Fe^3+^ ions, the dopa-modified gelatin forms a sticky hydrogel within seconds through complexation between the dopa molecules and Fe^3+^ ions [5]. In contrast to covalent bonds, metal–dopa bonds can spontaneously reform after breaking [6], which endows it with self-healing ability under certain conditions.

Dopa has a strong binding affinity to a variety of metal oxide surfaces due to three aspects: first, the di-hydroxy functional group of catechol enables it to form the stable bidentate modes of H-bonding with the surface of the oxide substrate [7]; second, the aromatic ring of catechol can form the strongest non-covalent cation–π interaction with positively charged metal ions [8,9]; third, catechol forms strong, reversible interfacial bonds with metal oxide surfaces [10,11,12,13]. Therefore, proteins and polymers containing catechol structure can form polymer metal complexes with Fe^3+^ ions, and can firmly adhere to the surfaces of metal oxides.

Inspired by this behavior, we designed a supramolecule end-capped with catechol that achieved high adhesion to metal substrates. The reaction was simply achieved through formation of a Schiff base between commercial PEA and a natural compound, 3,4-dihydroxybenzaldehyde, in ethanol. The resulting catechol-terminated PEA can feasibly perform complexation with Fe^3+^ ions, and the further evaporation of ethanol leads to clusters of coordination domains that serve as a physical cross-linking network. It is interesting to find that the interfacial coordinative interaction between the metal atoms of substrates (Al or Fe) and catechol or its oxidized form, i.e., quinone, is the overwhelming factor contributing to the improvement of adhesive force. This unique feature offers this supramolecular glue a bright potential in the application as an adhesive for metals.

## 2. Experimental

### 2.1. Materials

Poly (propylene glycol) bis (2-aminopropyl ether) (PEA, Mn = 2000 g/mol, f = 2) was purchased from Macklin. In addition, 3,4-dihydroxybenzaldehyde (dhba) was purchased from Bide. Fe (NO_3_)_3_·9H_2_O was purchased from Macklin. Ethanol was obtained from Titan and used without further purification.

### 2.2. Synthesis of Catechol-Terminated PEA

The synthesis is achieved through the following procedure: the reaction takes place between the catechol motif of dhba with the primary amine motif of PEA to generate the target polymer with a Schiff base and catechol groups located at the telechelic sites (Figure 1). Then, 2.76 g 3,4-dihydroxybenzaldehyde solid was dissolved in 30 mL ethanol, added to 20.00 g PEA, and reacted in a three-necked flask at room temperature for 24 h to obtain a brown-yellow solution.

### 2.3. Modification of Catechol-Terminated PEA with Fe^3+^

Catechol-terminated PEA was cross-linked with Fe^3+^, as shown in Figure 2. The complexed catechol-terminated PEA solution was used for subsequent experiments. Herein, four different polymer networks were prepared and denoted as PEA-dhba-Fe^3+^-15:1, PEA-dhba-Fe^3+^-10:1, PEA-dhba-Fe^3+^-5:1, and PEA-dhba-Fe^3+^-3:1, in which the mole ratio of catechol:Fe^3+^ equates to 15:1, 10:1, 5:1, and 3:1, respectively. As an example, to prepare the sample PEA-dhba-Fe^3+^-5:1, 1.61g Fe (NO_3_)_3_·9H_2_O dissolved in water was added to the above catechol-terminated PEA solution and reacted for 30 min to form a dark-brown solution. The solution was adjusted to a specific pH, poured into a Teflon dish, and allowed to evaporate at 80 °C to remove ethanol and water completely to obtain the cross-linked polymer network.

### 2.4. Adhesion Properties Tests

A tensile testing machine (TH-8100A, Tuobo, Suzhou, China) was used to carry out the tensile shear strength test. The test standard and sample preparation refer to GB/T 7124-2008, and the aluminum plate with a specification of 100 mm × 25 mm × 1.5 mm was used for lay-up. The lap area was set to 25 mm × 12.5 mm, and the number of samples brought to tests for each treatment condition was at least 5. They were heated in an oven at 80 °C for 8 h, and samples were conditioned at this temperature for 24 h prior to testing. Shear strength tests were performed with a constant speed of 50 mm/min at room temperature. The tensile shear strength τ = F/BL uses the unit of MPa, where F is the maximum load of specimen shear failure (N), B is the width of the lap surface (mm), and L is the length of the lap surface (mm). The film-like sample was placed on one end of the substrate, and then overlapped, fixed with a small clip, heated at different temperatures for different times, conditioned at room temperature to the equilibrium state, and then carried out for shear strength test.

### 2.5. Characterization and Measurements

Bulk samples were analyzed via FTIR analysis in attenuated total reflectance (ATR) mode with a machine obtained from Bruker ALPHA II (Baden-wuerttemberg, Germany). The ^1^H nuclear magnetic resonance (^1^HNMR) was recorded using a Bruker Advance 400 MHz (Baden-wuerttemberg, Germany) with deuterated DMSO as the solvent at room temperature. The PEA-dhba-Fe^3+^ cross-linking stoichiometry and the dhba oxidation were monitored on a UV–visible light spectrophotometer (Hewlett Packard 8453, G1103A, Palo Alto, CA, USA) using a quartz cuvette with a path length of 1 cm. Raman spectra were collected on a LabRAM HR Evo system equipped with an 800 nm laser line. SEM images and elemental maps of the cross-sections of the polymer networks were obtained using a Hitachi SU5000 SEM system. Rheological properties were determined by using a MCR302 rheometer. A sample (1 mL) or a circular film sample with a diameter of 20 mm and a thickness of approximately 1 mm, was poured into a holder, and the operation temperature was maintained at 37 °C. Elastic (G’) and loss moduli (G’’) were calculated from the frequency–modulus curves between 0.01 and 1.0 Hz. X-ray photoelectron spectroscopy (XPS) was performed using a Thermo SCIENTIFIC ESCALAB 250Xi K-alpha apparatus (hν (Al Kα) = 1486.6 eV). The pass energy was fixed at 30 eV. A Shirley background was systematically subtracted. Sputtering was achieved with Ar+ ions (500 eV, 10 mA, Raster area of 2 mm^2^) to remove the glue layer formed during transfer to the XPS apparatus. The first-principles density functional theory (DFT) calculations were performed using the Vienna abinitio simulation package (VASP) from the website (19 February 2022, https://www.materialsproject.org/materials/mp-990448/#). In our models, the Fe and Al surfaces (100) were adsorbed organic molecules for charge density calculations. Moreover, in order to obtain the accurate structures and adsorption energy of the Fe and Al surfaces, the calculations were performed using the hybrid functional as proposed by PBE interactions and are represented using the projector augmented wave (PAW) potential, and the Kohn–Sham one-electron valence states were expanded on the basis of plane waves with a cutoff energy of 400 eV. The Hellmann–Feynman forces convergence criterion was set as less than 0.05 eV Å^−1^, the K-point of 2 × 2 × 1 was used for the optimization of the Fe and Al surfaces (100), and the atomic layer with the thicknesses at about 12.0–17.0 Å was anchored.

## 3. Results and Discussion

### 3.1. Molecular Design

As shown in Figure 1a, PEA-dhba was synthesized via a one-pot method from commercially available and inexpensive raw materials. The dhba extracted from plant resources is chosen as the catechol-containing molecule for capping the PEA. The reaction is able to take place in ethanol at room temperature due to the high activity between the aliphatic primary amine and aromatic conjugated aldehyde. The resulting PEA-dhba is actually a telechelic polymer with dual terminal catechol motifs, which are functional groups that construct the supramolecular network by complexing with cations. The chemical shift of the hydrogen in benzaldehyde (-HC=O) performs upfield shift after reaction (Figure 1b), which indicates the successful formation of a Schiff base (-HC=N) in PEA-dhba. The conversion of PEA-dhba can also be revealed by FTIR spectra: both Figure 1c and Appendix A explicitly signify the existence of a stretching vibration of C=N at 1645 cm^−1^ for PEA-dhba and its complexed resultant PEA-dhba-Fe^3+^. Notably, the stretching vibration for quinone is apparent at 1688 cm^−1^. This implies the oxidation of catechol motifs in ethanol, occurring probably because of the dissolved oxygen. This transition of catechol is likely to influence the construction of the supramolecular network and its adhesive performance.

### 3.2. Characterizations

UV-vis spectroscopy was applied to gain insights into the formation of the supramolecular network and its complexation with Fe^3+^. UV-vis absorbance displays a strong peak at 395 nm, corroborating the existence of quinone motifs converted from catechols (Figure 2a). There are two observable peaks contributing to the complexation of catecholato-Fe^3+^. Theoretically, the maximal absorption values for individual mono-, bis-, and tris-catecholato-Fe^3+^ complexes are located at ~759 nm, ∼575 nm, and ∼492 nm, respectively [14,15,16,17,18]. In this case, the observable catecholato-Fe^3+^ peaks are located at ~501 nm and ~622 nm, which are likely caused by the overlapping of the three individual peak signals. The peak with stronger intensity at ~501 nm signifies that the tris-complex is predominant, whereas the bis-complex is in secondary dominance and the mono-complex is quite minor in complexation.

The supramolecular network was further analyzed via Raman spectroscopy (Figure 2b). The chelation of Fe^3+^ ions by the C3 and C4 oxygen atoms of catechol can be represented by a band consisting of peaks in the range of 490~694 cm^−1^, whereas the corresponding ring vibrations of catecholato-Fe^3+^complexes output peaks near 1323 cm^−1^ and 1484 cm^−1^. Specifically, the peaks at 587 cm^−1^ and 624 cm^−1^ are assigned to catecholato-Fe^3+^ complexation, and the other two peaks within this band are likely caused by the quinone-Fe^3+^ complexes. As expected, the peaks related to Fe^3+^-O chelation and ring vibration cannot be observed for PEA-dhba. However, apparent peaks are observed for PEA-dhba-Fe^3+^ with 1:3 and 1:5 feed ratios, indicating the formation of catechol-Fe^3+^ and quinone-Fe^3+^ coordination. In particular, the intensity of the peak at 587 cm^−1^ is much higher than that of the peak at 624 cm^−1^, indicating the dominance of tris-catechol-Fe^3+^ complexes in the polymer networks, which is in agreement with the UV-vis absorbance analysis. It can be also deduced from Figure 2a that the complexation modes remain nearly identical at pH = ~5 and pH = ~8, despite the fact that researcher-reported pH can have an intense effect on complexation modes [5,19,20]. This is presumably due to the inadequate amount of catechol motifs. The increase in the Raman intensity of the peaks at 587 cm^−1^, 1323 cm^−1^, and 1484 cm^−1^ for PEA-dhba-Fe^3+^ with 1:3 compared to 1:5 verifies this assumption.

It is universally acknowledged that the quinone form is likely to lead to coupling between catechol and quinone [1,4,5,21]; as a result, the oxidized PEA-dhba may experience chain extension. However, the GPC results show a nearly unchanged molecular weight and polydispersion index (Figure 2c), which is evidence against the chain extension thought to be caused by dopa-quinone coupling. To this end, the predominant chemistry of PEA-dhba and PEA-dhba-Fe^3+^ can be displayed in Figure 2d,e. The catechol motifs dangling in the chain ends are sensitive to oxidation, even with trace amount of oxygen in the ethanol. Nevertheless, further transition to coupled catechols nearly does not take place. Fe^3+^ ions are far more ready to perform complexation with catechol motifs, and the tris-complex is the dominant form, meaning that the individual PEA-dhba chain ends are assembled into supramolecular networks through coordination cross-linking. The SAXS results obtained from dried PEA-dhba-Fe^3+^ bulk show a conspicuous scattering peak and ring (Figure 2f). The inter-cluster spacing is calculated to be 3.70 nm, which can be considered to be phase separation between the soft PEA chains and coordination-induced clusters. It strongly confirms the formation of clusters of coordination that serve to construct physical cross-linking networks.

The chemical change of PEA-dhba and PEA-dhba-Fe^3+^ can be reflected by its color. As shown in Appendix A, the dark-brown solution of PEA-dhba is probably due to the formation of imine bonds. However, further addition of Fe^3+^ ions turns the solution even thicker and darker, which indicates the derivation of catechol into quinone and the formation of a catecholato-Fe^3+^ complex. This phenomenon was previously revealed by Marina Faiella et al., who showed that Fe^3+^ promotes the oxidization of catechols to quinones and mutually forms a dark-brown complex [22]. The change of rheological state is another sign of coordination. Unlike conventional waterborne adhesives, such as latex-containing high molecular weight epoxy, polyurethane, polyolefins, and so on [3,20,22], PEA-dhba-Fe^3+^ solutions in alcohols contain low molecular weight polymer chains. Whether or not the well-designed PEA-dhba-Fe^3+^, created according to the above protocols, could evolve into elastic supramolecular polymeric bulk has to be inspected in terms of rheological parameters. As shown in Figure 2g,h, the elastic and loss modulus of dried PEA-dhba-Fe^3+^ at 1.0 Hz are measured to be 28,587 Pa and 43,274 Pa, much higher than those of PEA-dhba (0.82864 Pa and 0.52505 Pa) and PEA (0.93545 Pa and 1.8709 Pa). The results indicate that the rapid cross-linking between catechol groups of PEA-dhba and Fe^3+^ ions significantly improves the mechanical properties. This is because the elastic modulus reflects the rigidity of the material, and the loss modulus represents the viscosity of the material. It is worth mentioning that the loss modulus of PEA-dhba-Fe^3+^ is obviously greater than the elastic modulus, which means that the viscosity of PEA-dhba-Fe^3+^ gives it its fundamental properties. This is because the catechol structure has strong adhesion. This phenomenon is also in line with our design. Appendix A shows the EDS mapping results of the cross-section of the PEA-dhba-Fe^3+^ bulk sample. It is clear that the C, N, O, and Fe elements are evenly distributed in the testing area, indicating the uniform structure of the polymer sample.

### 3.3. Adhesion Properties

This work develops a strategy for designing an environmentally friendly adhesive liquid polymer adhesive, which is composed of non-toxic polymeric components and diluent, non-covalent cross-linking networks that give it elasticity, non-reactive molecular motifs, and strong adhesion. Catechol motifs have the remarkable ability to adhere to a broad variety of metal and metal oxide surfaces, which makes it an ideal anchoring group for surface modification [2,3,4]. However, the binding strength to metal substrates is highly dependent on the number of its phenolic groups, which means its binding strength will drastically reduce when the catechol is oxidized to quinone and coordinated with the incorporated Fe^3+^ ions (Figure 2). Assuming that predominant catechols exist in their reduced form, the amount of residual phenolic groups in the four samples at different states is shown in Appendix A.

Lap-shear tensile strength tests were conducted by using dried PEA-dhba-Fe^3+^ films (Figure 3a) to adhere the aluminum substrates. By comparing the shear strength under different ratios and different pH values, it was found that the shear strength firstly increases and then decreases with the increase in the Fe^3+^:dhba ratio and pH. With a pH fixed at 5, the highest shear strength was 1.13 ± 0.15 MPa, as the ratio of Fe^3+^:dhba ratio was equal to 1:5 (Figure 3b,c). This ratio was then selected for the subsequent investigation of the adhesive properties. By adjusting the pH value of the precursor PEA-dhba-Fe^3+^ solutions, the highest shear strength reached 1.36 ± 0.14 MPa at pH = 8, which is approximate to that of seawater (Figure 3d and Appendix A). It explains that the cross-linking density of catechol-Fe^3+^ and the amount of residual -OH are both optimal for binding under mild conditions. The tested steel plates indicate the mechanical strength and shear strength of the bulk (Figure 3e). A low feed ratio and a high pH value lead to complete cohesive failure and low shear strength. The feed ratio at 1:5 and pH = 8 offer appropriate bulk mechanical strength that leads to the highest shear strength in spite of partial cohesive failure.

Then, we tested the shear strength of the lap-shear samples at 25 °C, 40 °C, 60 °C, and 80 °C for 15 min and 8 h, and the results are shown in Appendix A. When treated at 80 °C for 15 min and 8 h, the highest shear strength can reach 1.01 MPa and 1.38 MPa, respectively, indicating that catechol-terminated PEA with Fe^3+^ can be bonded very firmly, even in a short time. Bonding tests on different substrates show that the PEA-dhba-Fe^3+^ has strong bonding ability to aluminum and steel (Figure 3f,g), and minor adhesion to epoxy substrates, PET, and PVC. The emission of volatile organic compounds (VOCs) by adhesives is a major concern for consumers. Commercial adhesives developed with organic solvents or synthetic precursors release VOCs, which may cause undesirable skin and respiratory diseases. VOCs are released from commercial adhesives (i.e., the ERGO 0113 acrylic adhesive and the two-component epoxy adhesive) and PEA-dhba-Fe^3+^ in vessels. A gas chromatograph equipped with a mass detector (GC-MS) was used to monitor the number of instantly released VOCs over time. As shown in Figure 3h, areas of peak integration directly reflect the number of VOCs. It can be found that PEA-dhba-Fe^3+^ released very few VOCs within the first 25 min, whereas the epoxy adhesive had an abrupt emission of certain chemicals at ~10 min, and the acrylic adhesive constantly intensively emitted VOCs within the 25 min. On the basis of these comparative results, PEA-dhba-Fe^3+^ has significantly better environmentally friendly features. Hence, this liquid glue can be widely used in many industrial applications with consideration to its strong adhesion, low VOCs emission, and feasible operation.

The impressive adhesion to metal substrates in comparison to polymeric substrates indicates that PEA-dhba-Fe^3+^ might have unique interfacial interactions derived from its molecular structure. The mechanisms of formation of the metal/polymer interfaces depend on three main features: the availability of reactive functional groups at the polymer surface, the nano- and micro-structure of the surface, and the valence of the metal adsorbate(s). Metals that easily form oxides, such as Al, Cr, Fe, or Ni, will form stable metal (M-)O-C covalent bonds; for instance, Al atoms attack carbonyls and the in-chain C=O of polycarbonate to from C=O-Al [23,24,25]. It has been shown that the polymer/metal interface derived from coordination bonding, i.e., the electron donor (groups within organic components) and the electron acceptor (metal atoms or ions) produce a charge transfer to form an electric double layer, would produce desirable adhesion [26].

### 3.4. XPS Analysis and Theoretical Calculations

We performed XPS characterizations and theoretical calculations in order to figure out the science behind this phenomenon. The interfacial interaction between PEA-dhba-Fe^3+^ and metal substrates (i.e., iron and aluminum) can be qualitatively interpreted by the shift of binding energy in XPS spectra. This helps us to obtain a signal revealing the interface, which is revealed by evaporation and drying the adhesive in an oven. The signal was collected by electron-penetrating the coating layer for roughly 10 nm. Figure 4a compares the spectra of Al 2p for the substrate matrix and the interfacial layer. The peaks at 74.4 eV and 71.7 eV correspond to the 2p electrons of Al^3+^ in Al_2_O_3_ and Al atoms, respectively. The relatively stronger peak for Al^3+^ than that of Al is caused by the feasible oxygenation of the surface of the aluminum plate under an ambient atmosphere. It can be notably observed that a downshift of binding energies for both Al^3+^ and Al takes place when the excitation electron beam bombards the adhering surface. This is telling evidence of the formation of coordination bonds. Here, the oxygen-donating electron fills the empty orbital of Al^3+^ and Al. One more interesting phenomenon is that the peak intensity for Al←O surpassed that of Al^3+^←O, which indicates that the supramolecule of PEA-dhba-Fe^3+^ thoroughly wetted the substrate and can permeate into the bulk Al. Likewise, the spectra of Fe 2p for the substrate matrix and the interfacial layer are shown in Figure 4b. The Fe 2p core peaks for the bulk steel sample are split into two components because of spin-orbit coupling (Fe 2p_3/2_ and Fe 2p_1/2_) with an area ratio of 2 and present multiplet structures and satellites [27]. It can also be observed that the peaks for 2p_1/2_ and 2p_3/2_ undergo obvious downshifts, giving them lower binding energies when they adhere to the interface sample, which also corroborates the formation of coordination bonds (Fe←O) between the steel substrate and PEA-dhba-Fe^3+^. The O 1s spectrum is further deconvoluted in order to reveal the exact oxygen-containing groups involving the coordination. High-resolution scans of the O 1s for the drop-coated aluminum substrate (Figure 4c) show the presence of C_Ph_-OH/C-O-C and (-C=O)_quinone_ functional groups with deconvoluted binding energies at 533.0 eV and 531.4 eV, respectively. There are two peaks occurring at 533.8 eV and 532.2 eV when adjusting the cumulative fit to overlap the raw curve. They are attributed to the complexed form of C_Ph_-OH/C-O-C and (-C=O)_quinone_, namely, C_Ph_-O-Al/C-O-C and (-C=O-Al)_quinone_. A similar process happens to the drop-coated steel substrate (Figure 4d), i.e., the peaks for C_Ph_-OH/C-O-C and (-C=O)_quinone_ at 532.5 eV and 530.7 eV undergo an upshift of binding energies to 533.5 eV and 531.5 eV. To this end, the XPS results firmly demonstrate the existence of coordinative interactions at the adhering interfacial layer. The interfacial complexation serves to generate overwhelming adhesive force compared to other polymeric substrates.

It is revealed by the quantum simulation results that the quinone and catechol motifs spontaneously perform strong adsorption toward Al or Fe atoms. In this study, density functional theory (DFT) was used to calculate the electronic property and adsorption energy of Fe and Al (100) surface-adsorbed organic molecules (see supplementary chemical formula). According to the calculation of the adsorption energy, the adsorption energy of organic molecules to Al and Fe is −2.227 eV (Figure 5a) and −1.439 eV (Figure 5g), respectively. According to the bond energy manual, when the absolute value of the energy is higher than 1eV, the interaction is a strong interaction, namely, chemical adsorption occurs instead of a van der Waals interaction. Therefore, formation bonds take place between organic molecules and metals, which greatly contributes to the notable improvement of the adhesion to metal substrates. The charge density of the organic molecules adsorbed on the metal surface is shown in Figure 5b,h. The yellow cloud symbolizes the area of withdrawing electrons, and the blue cloud symbolizes the area of donating electrons. It can be seen from the figures that both the Al and Fe atoms lose electrons after adsorbing organic molecules, while the counterpart organic molecules gain electrons. The transfer of electrons from metal to organic molecules is further demonstrated by the cross-sectional analysis of the charge density (Figure 5c,i) in the adsorption region, where the red region is electron-enriched, while the blue region is the electron-poor region, further indicating that the electrons are transferred from metal to organic molecules. In addition, we calculated the electron localization function (ELF) and electrostatic potential distribution for the adsorption of organic molecules on the Al and Fe metal surfaces. As the redder color indicates a higher degree of localization, it is found that the degree of delocalization of electrons on the metal surfaces is significantly higher than that of the organic molecules (Figure 5e,j), which further indicates the aggregation of electrons on the surface of organic molecules. The electrostatic potential distribution proves that the electrostatic potential of the metal surfaces is higher than that of the organic molecules (Figure 5f,k). Regarding to the above analysis, the adsorption of organic molecules on the Al or Fe surfaces leads to the transfer of electrons from the metal to the organic molecules, and there is an obvious gain and loss of electrons at the adsorption site, which indicates that the metal and organic molecules form a strong ionic bond interaction.

According to the above results, a model revealing this environmentally friendly supramolecular glue can be explicitly demonstrated by the schematic illustration in Figure 6. As for the PEA-dhba-Fe^3+^ solution in ethanol, the amount of Fe^3+^ ions is notably insufficient to perform complexation with all catechol and quinone groups, meaning that the system remains as a fluidic liquid, except that its viscosity apparently increases. After the ethanol is removed, this can result in a dark-brown elastomer, which is the consequence of the clusters formed by the complexation of Fe^3+^ with catechol and quinone groups (Figure 2f). The clusters act as a cross-linking network within the bulk and, as a result, the material exhibits the features of an elastomer. One more peculiar merit of this elastomer is that it can be feasibly processed in molten state or re-dissolved in alcohol solvents. As for the origin of adhesion to metal substrates, the abundance of remaining catechol and quinone groups plays a pivotal role in the construction of a polymer/metal interface with coordination bonds. Even though the transition of catechol moieties into quinone is unavoidable due to oxygenation, the resultant quinone groups do not have negative impacts on adhesion, but rather they readily perform interfacial coordination bonds, a kind of strong interaction that contributes to substantially improving the shear adhesive strength with reference to the conventional van der Waals interactions.

## 4. Conclusions

In summary, we have designed a novel adhesive supramolecular glue with extraordinary binding properties for aluminum and steel plates. The catechol and quinone groups play a pivotal role in the formation of interfacial coordination bonds, contributing to the remarkable improvement of adhesion. Specifically, the shear strength for an aluminum plate can reach as high as 1.36 ± 0.14 MPa when the pH is adjusted to 8. This is an improvement over those adhered to polymeric substrates. The XPS qualitatively revealed the successful formation of coordination bonds, and the theoretical calculations simulated the adsorption energies between atoms of Al or Fe and the carbonyl oxygen of quinone, which substantially confirmed the chemical interfacial bonding and also quantitatively validated its overwhelming binding effect, leading to the absence of van der Waals interactions. In addition, PEA-dhba-Fe^3+^ contains no toxic solvents and emits minor VOCs with reference to commercial epoxy and acrylic adhesives; thus, it is highly promising that this environmentally friendly supramolecular glue can be used in cutting-edge applications as an adhesive for metal substrates.

## Figures and Tables

**Figure 1 polymers-14-00916-f001:**
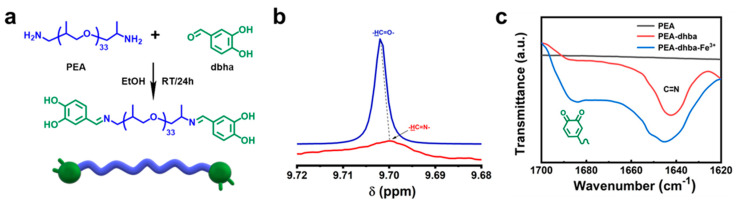
(**a**) Synthetic routes of PEA-dhba; (**b**) ^1^HNMR spectra of dhba and PEA-dhba-Fe^3+^; (**c**) FTIR spectra of PEA, PEA-dhba, and PEA-dhba-Fe^3+^.

**Figure 2 polymers-14-00916-f002:**
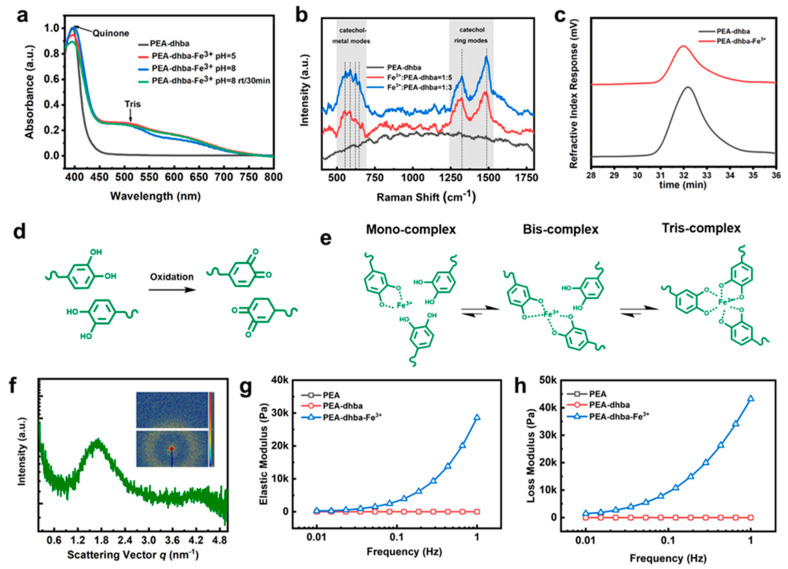
(**a**) UV-vis absorbance of diluted PEA-dhba and PEA-dhba-Fe^3+^ solutions; (**b**) Raman spectroscopy of neat PEA-dhba and complexes with 1:3 and 1:5 molar ratio; (**c**) GPC information of as-synthesized PEA-dhba and sample open to air for 6 h; (**d**) oxidation of catechols into quinone form; (**e**) complexation modes of catechol-Fe^3+^ in PEA-dhba-Fe^3+^; (**f**) SAXS results of dried PEA-dhba-Fe^3+^ bulk, and the inset shows its two-dimensional (2D) pattern; (**g**) elastic modulus; and (**h**) loss modulus obtained from frequency-sweeping rheological measurements for dried PEA, PEA-dhba, and PEA-dhba-Fe^3+^.

**Figure 3 polymers-14-00916-f003:**
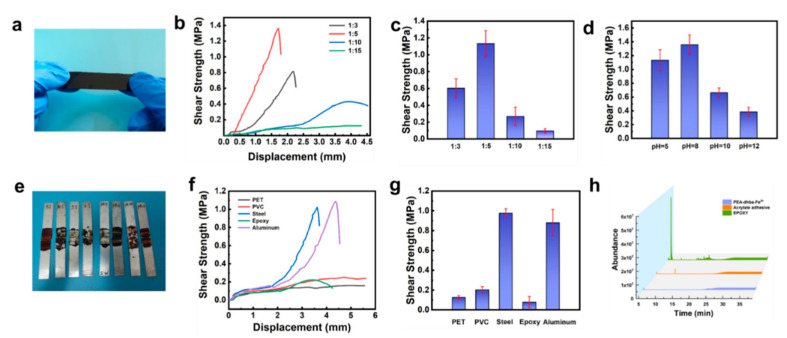
(**a**) Photograph of PEA-dhba-Fe^3+^ film; (**b**) shear strength plotted against displacement of PEA-dhba with various ratios of catechol:Fe^3+^ and (**c**) corresponding shear strength comparison; shear strength of samples adhered to aluminum by pasting PEA-dhba-Fe^3+^ with various (**c**) feed ratios of Fe^3+^:dhba and (**d**) pH values; (**e**) photographs showing stretched lap-shear samples; (**f**) shear strength plotted against displacement of PEA-dhba-Fe^3+^ on various substratum and (**g**) corresponding shear strength comparison; (**h**) GC-MS chromatograms for VOCs released from PEA-dhba-Fe^3+^, acrylic, and epoxy adhesives.

**Figure 4 polymers-14-00916-f004:**
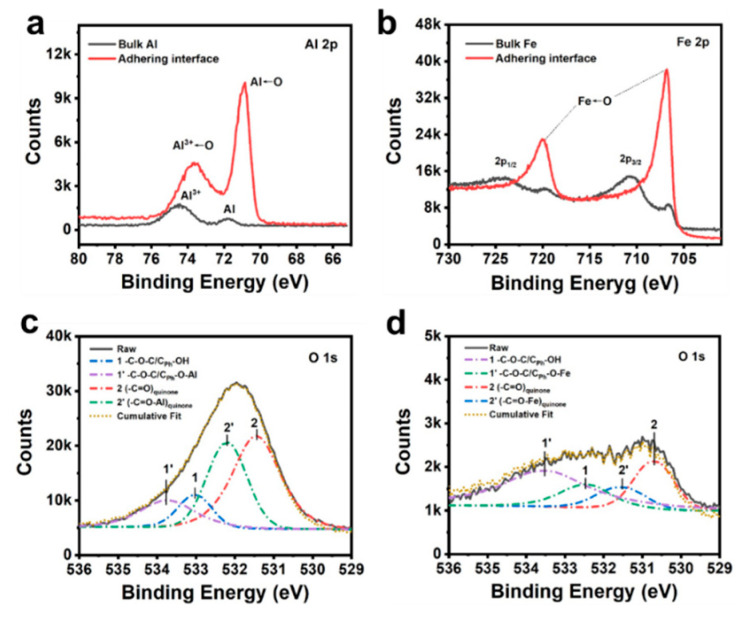
(**a**) Al 2p and (**b**) Fe 2p XPS spectra for the bulk substrate and the adhering interface; deconvoluted O 1s XPS spectra for (**c**) Al-adhering interface and (**d**) Fe-adhering interface.

**Figure 5 polymers-14-00916-f005:**
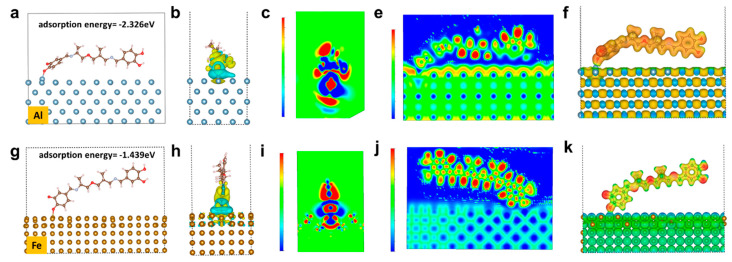
Electronic property and adsorption energy for (**a**) Al and (**g**) Fe; charge density for organic molecule adsorbed on (**b**) Al and (**h**) Fe; cross-sectional analysis of charge density for organic molecule adsorbed on (**c**) Al and (**i**) Fe; electron localization function for organic molecule adsorbed on (**e**) Al and (**j**) Fe; electrostatic potential distribution for organic molecule adsorbed on (**f**) Al and (**k**) Fe.

**Figure 6 polymers-14-00916-f006:**
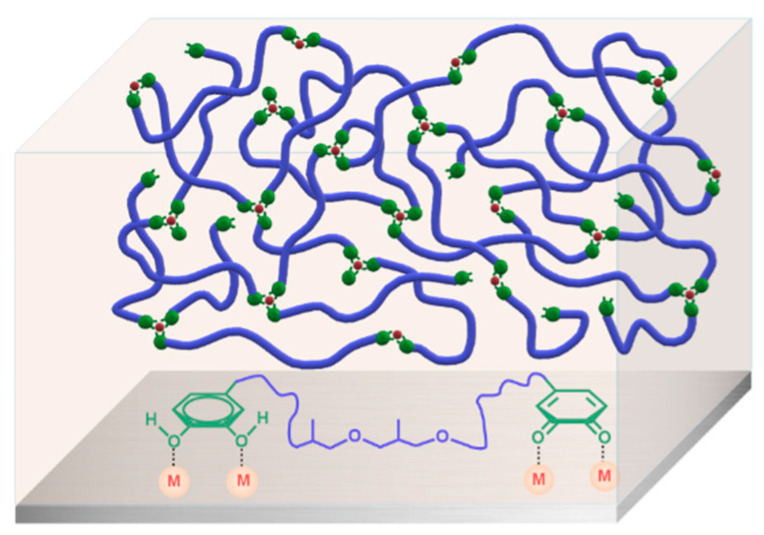
Schematic illustration of the supramolecular network of PEA-dhba-Fe^3+^ and interfacial interactions between the residual catechol and quinone groups and metal substrate.

## Data Availability

Not applicable.

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
