# Peer review of "An Environmentally Friendly Supramolecular Glue Developed from Natural 3,4-Dihydroxybenzaldehyde"

_polymers, 2022, doi:10.3390/polym14050916_

Round 1
Reviewer 1 Report
The manuscript reports on the development of a new glue media derived from polyetheramine, 3,4-dihydroxybenzaldehyde and Fe3+ salt. The properties of the resulting matter were characterized by several physical methods. The findings of the authors are mostly convincingly confirmed. The paper can be published after some improvement.
1. The English is quite satisfactory, but some flaws should be corrected. For example, the sentence "The tri-complex is found be..." (Line 15) is problematic. The strategy hereby for designing... (line 218) - a verb is lost.
My main remarks concern the chemistry of the described processes.
2. The authors state: "As is universally acknowledged that the quinone form is likely to perform inter-aroma coupling [22,24-26], as a result, the oxidized PEA-dhba may lead to chain extension." (Line 185) What is this "inter-aroma coupling"? It is not mentioned in Refs. [22,24-26]. Ortho-benzoquinone undergoes Diels-Alder reaction with itself, perhaps this was meant, and "coupled catechol" is a Diels-Alder adduct? This should be clarified.
Also, conserning this point, "aroma" means aromatic? I doubt such an abbreviation exists in chemistry.
3. Between catecholes and their fully oxidized relatives, o-quinones, there are intermediate o-semiquinone radical anions that form well-known dark-colored complexes with transition metals, with iron 3+ as well (see, for example [Robert M. Buchanan et al., JACS, 1978, 100, 25, 7894, DOI:10.1021/ja00493a018]). Fe3+ promotes the oxidation of catecholes to semiquinones as an easily and reversibly reducible oxidizer. It is highly likely that most of the iron in the substance is present in the form of semiquinone complexes, which can be clarified by EPR study. This question should be specified.
Also, spectroscopy data should be reviewed due to the possible presence of semiquinone complexes. It is absolutely impossible that catechols and quinones exist in the same system, but there would be no semiquinones in it.
After clarifying this issues, in my opinion, the article can be published.
Author Response
Dear reviewer, I have attached the specific reply to the document below.

Reviewer 2 Report
The manuscript by Wang et al. reports the synthesis of PEA-ahba-Fe composite, which was tested as adhesives. The manuscript is written fairly well and fits the scope of the Journal.
Some remarks for the authors to consider:
- In Figure 1c, FT-IR spectra are shown in the range 1700-1620 cm-1. It would be informative to also show the change in the area around 3500 cm-1, where OH stretching vibration bands would appear.
- In Section 3.4, quantum-chemical simulations are mentioned and some of their graphic representations are shown in Figure 4e,f. However, no computational details are given in the text.
- Throughout the text, there are references to Figures and Tables in the Supporting information, however, this information in absent in the submission.
Author Response

(The authors gave the same response as above.)

Reviewer 3 Report
The submitted paper entitled “An environmental-friendly supramolecular glue developed from natural 3,4-dihydroxybenzaldehyde” is an interesting manuscript concerning the synthesis and characterization of novel polymeric material with possible application as non-toxic glue for example for metal surfaces. Authors applied nonobvious approach employing coordination bonds to improve the adhesive properties of a polymer.
The manuscript is well written, the experiments were well planned and the conclusions seem to follow the results. I can recommend this article to be published in Polymers, however I have some minor remarks.
- How the authors chose the proper ratio of PEA and 3,4-dihydroxybenzaldehyde? Was there any rational reason to apply such proportion (2.76g to 20g)? Did the authors make some other trials? Was it optimal? What were the other results?
- The authors performed many experiments to characterize the novel material. In my opinion it would be interesting to retrieve even more information from the presented data. For example the authors should discuss the size of clusters obtained from SAXS method. Did the authors perform the SAXS experiments on PEA and PEA-dhba? Were there any conspicuous differences?
- Please correct the typo at p. 9 “sur-pmolecular”
- Please unify the abbreviation of 3,4-dihydroxybenzaldehyde “dbha” or “dhba”?
Author Response

(The authors gave the same response as above.)
